

# Macroscopic fluctuation theory and current fluctuations in active lattice gases

Tal Agranov[1], Sunghan Ro[2], Yariv Kafri[2] and Vivien Lecomte[3]

**1** Department of Applied Mathematics and Theoretical Physics, University of Cambridge, Wilberforce Road, Cambridge CB3 0WA, United Kingdom.
**2** Department of Physics, Technion-Israel Institute of Technology, Haifa, 3200003, Israel.
**3** Université Grenoble Alpes, CNRS, LIPhy, 38000 Grenoble, France.

## Abstract

We study the current large deviations for a lattice model of interacting active particles displaying a motility-induced phase separation (MIPS). To do this, we first derive the exact fluctuating hydrodynamics of the model in the large system limit. On top of the usual Gaussian noise terms the theory also presents Poissonian noise terms, that we fully account for. We find a dynamical phase transition between flat density profiles and sharply phase-separated traveling waves, and we derive the associated phase diagram together with the large deviation function for all phases, including the one displaying MIPS. We show how the results can be obtained using methods similar to those of equilibrium phase separation, in spite of the nonequilibrium nature of the problem.

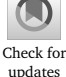

# 1  Introduction

The modeling of active matter has attracted a lot of attention in recent years [1–15]. This is due to the many novel phenomena exhibited by active systems that are not seen in equilibrium. The macroscopic description of active systems is based, in many cases, on noiseless phenomenological field theories, mean-field approximations or gradient expansions [16–25]. Noise is typically added in as Gaussian contributions, with limited control. Understanding the nature of noise in a more rigorous way is becoming important with the growing interest in fluctuations in active systems [26–38]. Recently, an active lattice gas model with an exact hydrodynamic description has been introduced in Refs. [39,40]. While the model presents a diffusive scaling, it reproduces characteristic features of scalar active systems, namely, motility-induced phase separation (MIPS) [41,42]. Interestingly, this phase transition already appears in one dimension.

In a previous work [43], we derived the fluctuating hydrodynamics of this model in the regime of small typical Gaussian fluctuations. This enabled us to find the static and dynamical correlation functions of the model in the homogeneous phase. In particular, close to the critical point, we showed that the scaling exponents belong to the Ising mean-field universality class.

It is also of interest to understand large fluctuations in such models beyond the typical regime which is captured by small Gaussian fluctuations. These are captured by large-deviation theory and have been studied extensively in non-equilibrium systems (see for instance Refs. [44–47]). Much work has focused on models which admit a diffusive scaling between space and time coordinates when taking the large system size limit [48,49]. In this limit the noise is small, and in many cases both typical and atypical fluctuations are Gaussian. Then the fluctuations are fully described by the Macroscopic Fluctuation Theory (MFT) (see [50] for a review). This framework allows one to successfully describe the distribution of several observables of interest, such as the density profile or the time-integrated current flowing through the system [45,51–53]. One of the most interesting features exhibited by the fluctuations are dynamical phase transitions, which occur when the large-deviation function is singular [52,54–61]

However, systems in which particles not only diffuse by jumps but also undergo reactions evade a Gaussian MFT description, as is was found in equilibrium [62] or driven [63] models.

Although the scalings in such systems are still diffusive, one has to take into account the Poissonian nature of the noise if one wants to fully describe their fluctuations. The active model we are interested in falls in that class: it consists of left- and right-moving particles which jump but also tumble. The latter is represented by a reaction where one type of particle transforms into the other.

In this paper, we derive a non-Gaussian MFT that fully encompasses the Poissonian noise, extending our previous work on small Gaussian fluctuations [43]. We then apply this extended MFT framework to study the distribution of the time-integrated current flowing through the system. These were studied previously analytically only for non interacting particles [35] or numerically in the interacting case [27].

We find that current fluctuations in the active lattice gas model exhibits a dynamical phase transition (DPT) and we explain how it can be derived using techniques very similar to those used in equilibrium phase transitions. The DPT occurs between a homogeneous profile and a sharply phase separated travelling wave. We also study in detail how this transition connects to the zero current phase diagram which exhibits MIPS. To do this we account for finite system lengths.

The structure of the paper is as follows: the model and its hydrodynamics are described in Sec. 2. The fluctuating hydrodynamics (beyond the Gaussian regime) is derived in Sec. 3. In Sec. 4, we obtain the corresponding MFT of the active lattice gas and use it to study the current large deviations. The dynamical phase transition presented by the system is studied in Sec. 5. Sec. 6 establishes the relation between the MIPS and the DPT. We conclude in Sec. 7. Technical steps of our derivations are gathered in appendices.

## 2 The active lattice gas model and its hydrodynamics

The active lattice gas model, introduced in Ref. [39], is defined on a one-dimensional periodic lattice with $L$ sites. Each site $i$ can be in either one of three states: occupied by a + particle, occupied by a − particle, or empty. The dynamics is defined through the following rates (see Fig. 1):

(i) A pair of neighboring sites exchange their states (if different) with rate $D_0$.

(ii) A + (−) particle hops using self-propulsion to the right (left) neighboring site with rate $\lambda/L$, provided that the target site is empty.

(iii) A + (−) particle tumbles into a − (+) particle with rate $\gamma/L^2$.

The scaling of the rates with $L$ ensures that in the hydrodynamic limit ($L \gg 1$), all processes occur on diffusive time scales. Indeed, the time it takes for particles to travel across $L$ sites, either through diffusive motion or using self-propulsion, scales as $L^2$ which is also the time scale for tumbling events.

The hydrodynamic equations of the model are obtained by defining the coarse-grained particle density fields

$$\rho_\pm(x,t) = \frac{1}{2L^\delta} \sum_{|i-Lx|<L^\delta} \sigma_i^\pm, \tag{1}$$

with $\sigma_i^+ = 1$ ($\sigma_i^- = 1$) if site $i$ is occupied by a + (−) particle and $\sigma_i^+ = 0$ ($\sigma_i^- = 0$) otherwise. The exponent $0 < \delta < 1$ defines a mesoscopic length, $L^\delta$, that scales sub-linearly with the system size. In what follows it is useful to use the rescalings $t \to \gamma t/L^2$ and $x = \ell_s i/L$, where $\ell_s^{-1} \equiv \sqrt{D_0/\gamma}$ is the typical distance traveled by a particle using only diffusive steps until it tumbles. Note that $x \in [0, \ell_s]$ so that $\ell_s$ plays the role of system length for the macroscopic

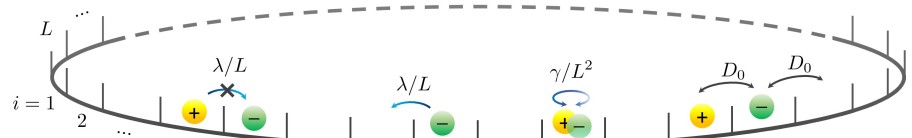

Figure 1: Schematic representation of the microscopic dynamics of the active lattice gas model. Particles self-propel with rate $\lambda/L$ to vacant sites and tumble at rate $\gamma/L^2$. They also perform diffusive swap with an occupied or vacant neighbor at equal rate $D_0$.

coordinate $x$. In Refs. [39,43] it was shown that the density fields defined in Eq. (1) obey the fluctuating hydrodynamic equations

$$\partial_t \rho_+ = -\partial_x J_+ - K, \qquad \partial_t \rho_- = -\partial_x J_- + K. \tag{2}$$

Here $J_+$ and $J_-$ are conservative density fluxes that arise due to the diffusion and self-propulsion. Their deterministic components are given by the expressions

$$\bar{J}_+ = -\partial_x \rho_+ + \text{Pe}\,\rho_+(1-\rho), \qquad \bar{J}_- = -\partial_x \rho_- - \text{Pe}\,\rho_-(1-\rho), \tag{3}$$

where $\rho(x,t) = \rho_+(x,t) + \rho_-(x,t)$ is the total density of particles and $\text{Pe} = \lambda/\sqrt{\gamma D_0}$ is the Péclet number, which compares the persistence length $\lambda/\gamma$ to the diffusive one $\ell_s^{-1}$. The non-conservative term $K$ arises from tumbling. It is the local rate at which $+$ particles tumble into $-$, minus the rate of the opposite reaction. Its deterministic component is given by

$$\bar{K} = \rho_+ - \rho_-. \tag{4}$$

For our purposes, it is more convenient to use the density and polarization fields,

$$\rho(x,t) = \rho_+ + \rho_-, \qquad m(x,t) = \rho_+ - \rho_-, \tag{5}$$

respectively, which follow the dynamics

$$\begin{aligned}
\partial_t \rho &= -\partial_x J_\rho, \\
\partial_t m &= -\partial_x J_m - 2K,
\end{aligned} \tag{6}$$

where the density and polarization fluxes are defined as $J_\rho \equiv J_+ + J_-$ and $J_m \equiv J_+ - J_-$. Correspondingly, the deterministic components of the conservative fluxes and tumbling rate in these variables are given by

$$\bar{J}_\rho = -\partial_x \rho + \text{Pe}\,m(1-\rho), \qquad \bar{J}_m = -\partial_x m + \text{Pe}\,\rho(1-\rho), \qquad \bar{K} = m. \tag{7}$$

In the $L \to \infty$ limit, the fluctuations are suppressed and Eq. (6) reduces to its deterministic hydrodynamics form given by $J_\rho$, $J_m$, and $K$ replaced by their average values given in Eq. (7). In this limit, the hydrodynamics predicts that the system relaxes to a steady state given by the stationary solutions of Eq. (6).

In Ref. [39], it was shown that for large enough Pe and mean density $\rho_0 = \ell_s^{-1} \int_0^{\ell_s} \rho(x,t)dx$, the equations allow for non-homogeneous stationary solutions corresponding to motility-induced phase-separation (MIPS). These solutions consist of coexisting high- and low-density phases separated by sharp domain walls, with the ratio between the system size and domain wall width controlled by $\ell_s$. In the $\ell_s \gg 1$ asymptotics, the densities in the phase-separated state are independent of the mean density $\rho_0$. They can be found using an effective common

tangent construction, see refs. [20,21]. The resulting phase diagram, obtained in Ref. [39], is shown in Fig. 2. It shares similarities with equilibrium phase separation, despite the very different origin of the two phenomena [42]. The figure also shows the spinodal line, determined by the equation $2 - \mathrm{Pe}^2(1 - \rho_0)(2\rho_0 - 1) = 0$, beyond which the homogeneous state becomes linearly unstable. The contact point of the MIPS line and the spinodal line defines the MIPS critical point $(\rho_c, \mathrm{Pe}_c) = (3/4, 4)$.

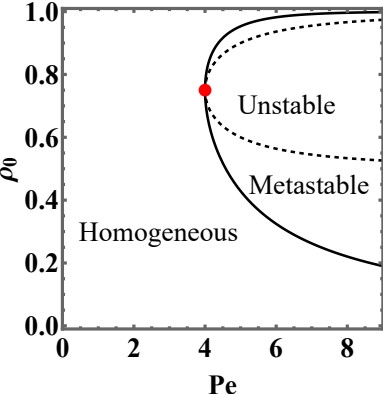

Figure 2: The phase diagram of the model. The binodal is denoted by a solid black line and the spinodal by a dashed line. They meet at $(\rho_0, \mathrm{Pe}) = (3/4, 4)$ which marks the critical point (red circle). In a MIPS state, the high- and low-density coexisting phases lay on the binodal.

# 3 Fluctuating Hydrodynamics

For systems with finite $L$, the fluctuations cannot be neglected anymore and the full statistics of $J_\rho$, $J_m$, and $K$ has to be taken into account.

On general grounds the fluctuations scale as $L^{-1/2}$ at large $L$. In Ref. [43], we exploited an exact mapping to the well studied ABC lattice gas model [64–66] to derive an expression for typical small Gaussian fluctuations. Using these we were able to characterize the critical behavior of the model exactly, finding that it belongs to the mean-field Ising universality class both for static and dynamic properties.

In this paper, we extend the above results to account for arbitrary large macroscopic fluctuations, beyond the Gaussian small-fluctuations regime. As we show, the fluctuations of the conserved fluxes $J_\rho$ around $\bar{J}_\rho$ and $J_m$ around $\bar{J}_m$ are described by Gaussian noise terms even for large fluctuations. This is the standard case in related lattice gases which are in local equilibrium [67]. The reason is that the fluxes are averaged over fast diffusive exchanges with a rate $D_0$ that does not scale with $L$ [68]. In contrast, the large fluctuations of $K$ around $\bar{K}$ are not Gaussian. They come from slow local tumbling events which follow Poisson statistics. The technical steps of the derivation of these results are presented in Appendix A.

The joint statistics of fluxes and tumbling rate fluctuations provide a complete statistical description of macroscopic fluctuations in the active lattice gas model, which is the first main result of this paper. It is given in terms of the probability path measure, $P$, of observing a history of the fields $\rho(x,t)$, $m(x,t)$, $J_{\rho,m}(x,t)$, $K(x,t)$ which, at large $L$, follows the large-deviation

form

$$-\ln P\left[\rho,m,J_{\rho,m},K\right] \simeq L\ell_s^{-1}\hat{\mathcal{S}}_{\mathcal{L}}\left[\rho,m,J_{\rho,m},K\right],$$

$$\hat{\mathcal{S}}_{\mathcal{L}}\left[\rho,m,J_{\rho,m},K\right] = \int_0^T dt \int_0^{\ell_s} dx \left(\mathcal{L}_J + \mathcal{L}_K\right). \tag{8}$$

Here $\mathcal{L}_J$ accounts for the statistics of the fluxes and is given by the quadratic form

$$\mathcal{L}_J = \frac{1}{2}\begin{bmatrix} J_\rho - \bar{J}_\rho \\ J_m - \bar{J}_m \end{bmatrix}^{\mathrm{T}} \mathbf{C}^{-1} \begin{bmatrix} J_\rho - \bar{J}_\rho \\ J_m - \bar{J}_m \end{bmatrix}, \tag{9}$$

with the correlation matrix $\mathbf{C}$ given by

$$\mathbf{C} = \begin{bmatrix} \sigma_\rho & \sigma_{\rho,m} \\ \sigma_{\rho,m} & \sigma_m \end{bmatrix}, \tag{10}$$

with entries

$$\sigma_\rho = 2\rho(1-\rho), \quad \sigma_m = 2\left(\rho - m^2\right), \quad \sigma_{\rho,m} = 2m(1-\rho).$$

The $\mathcal{L}_K$ term in the action, accounting for the Poisson tumble statistics is given by:

$$\mathcal{L}_K = \rho - \sqrt{K^2 + (\rho^2 - m^2)} + K\ln\left[\frac{\sqrt{K^2 + (\rho^2 - m^2)} + K}{(\rho + m)}\right]. \tag{11}$$

The action above is presented in its Lagrangian form. As usual in MFT, the fields $\rho(x,t)$, $m(x,t)$ and the currents $J_{\rho,m}(x,t)$, $K(x,t)$ are imposed, in the path measure, to satisfy conservation equations given by Eqs. (6). We now turn to use these expressions to derive the LDF for the total integrated current flowing in the system using the macroscopic fluctuation theory.

## 4 Macroscopic Fluctuation Theory

The total integrated current flowing through the system in a time interval $T$ is given by

$$Q = \int_0^{\ell_s} dx \int_0^T dt\, J_\rho. \tag{12}$$

At long times $T \gg 1$ and for large number of sites $L \gg 1$, the distribution of $Q$ takes the large-deviation form

$$-\ln P(Q) \simeq LTI(q), \tag{13}$$

where

$$q = \frac{Q}{T\ell_s}, \tag{14}$$

is a rescaled integrated current. As the system is reflection symmetric we have $I(q) = I(-q)$. For simplicity we will consider from now on only $q > 0$. To find the rate function $I(q)$, we first determine the scaled cumulant generating function (SCGF)

$$\psi(\Lambda) = \frac{1}{LT}\ln\langle e^{\Lambda LTq}\rangle. \tag{15}$$

Similar to equilibrium thermodynamics, following Varadhan's lemma [46], the rate function $I$ is related to the SCGF by a Legendre–Fenchel transform,

$$I(q) = \sup_{\Lambda}\left[\Lambda q - \Psi(\Lambda)\right]. \tag{16}$$

The SCGF can be expressed via a path-integral formulation :

$$e^{LT\Psi(\Lambda)} = \int \mathcal{D}\rho\, \mathcal{D}m\, \mathcal{D}J_\rho\, \mathcal{D}J_m\, \mathcal{D}K\, e^{L\ell_s^{-1}(\Lambda Q - \hat{\mathcal{S}}_\mathcal{L})}\, \delta(\dot\rho + \partial_x J_\rho)\, \delta(\dot m + \partial_x J_m + 2K). \qquad (17)$$

The delta functions ensure that the dynamical equations (6) are satisfied at each point of space and time. In the large-$L$ limit, $\Psi(\Lambda)$ can be evaluated using saddle-point asymptotics: $\Psi(\Lambda) = -\frac{1}{\ell_s T} \min_{\rho,m,\hat{p}_\rho,p_m} \mathcal{S}_{\text{tot}}$. Using standard techniques this translates to minimizing an action given by

$$
\begin{aligned}
\mathcal{S}_{\text{tot}} &= \int_0^{\ell_s} dx \int_0^T dt \left\{ \dot\rho \hat{p}_\rho + \dot m p_m - \frac{1}{2} \begin{bmatrix} \partial_x \hat{p}_\rho + \Lambda \\ \partial_x p_m \end{bmatrix}^T \mathbf{C} \begin{bmatrix} \partial_x \hat{p}_\rho + \Lambda \\ \partial_x p_m \end{bmatrix} - \bar{J}_\rho(\partial_x \hat{p}_\rho + \Lambda) \right. \\
&\left. - \bar{J}_m \partial_x p_m \right\} + \int_0^{\ell_s} dx \int_0^T dt \left( -2\rho \sinh^2 p_m + m \sinh 2p_m \right),
\end{aligned} \qquad (18)
$$

that we derive in Appendix B. Here $\hat{p}_\rho$ and $p_m$ are auxiliary fields, periodic in the spatial direction, introduced by writing the delta-function constraints in (17) using a Fourier representation. Introducing $p_\rho \equiv \hat{p}_\rho + \Lambda x$ the action takes the form

$$\mathcal{S}_{\text{tot}} = \int_0^{\ell_s} dx \int_0^T dt \left\{ \dot\rho p_\rho + \dot m p_m - \mathcal{H}[\rho, m, p_\rho, p_m] - \Lambda x \dot\rho \right\}, \qquad (19)$$

with the Hamiltonian density

$$\mathcal{H}[\rho, m, p_\rho, p_m] = \frac{1}{2} \begin{bmatrix} \partial_x p_\rho \\ \partial_x p_m \end{bmatrix}^T \mathbf{C} \begin{bmatrix} \partial_x p_\rho \\ \partial_x p_m \end{bmatrix} + \bar{J}_\rho \partial_x p_\rho + \bar{J}_m \partial_x p_m + 2\rho \sinh^2 p_m - m \sinh 2p_m. \quad (20)$$

Note that $p_\rho$ has a jump discontinuity of size $\Lambda \ell_s$ at $x = \ell_s$ on the ring geometry studied here: $p_\rho(\ell_s) = p_\rho(0) + \Lambda \ell_s$.

The optimal trajectories which minimize the action are solutions of the Hamiltonian MFT equations

$$
\begin{aligned}
\partial_t \rho &= \frac{\delta \mathcal{H}}{\delta p_\rho} = -\partial_x \left[ \sigma_\rho \partial_x p_\rho + \sigma_{\rho,m} \partial_x p_m + \bar{J}_\rho \right], \qquad &(21) \\
\partial_t p_\rho &= -\frac{\delta \mathcal{H}}{\delta \rho} = -\partial_x^2 p_\rho - (1 - 2\rho)(\partial_x p_\rho)^2 + 2m \partial_x p_\rho \partial_x p_m \\
&\quad - (\partial_x p_m)^2 + \text{Pe}\, m \partial_x p_\rho - \text{Pe}(1 - 2\rho)\partial_x p_m - 2\sinh^2 p_m, \\
\partial_t m &= \frac{\delta \mathcal{H}}{\delta p_m} = -\partial_x \left[ \bar{J}_m + \sigma_m \partial_x p_m + \sigma_{\rho,m} \partial_x p_\rho \right] + 2(\rho \sinh 2p_m - m \cosh 2p_m), \\
\partial_t p_m &= -\frac{\delta \mathcal{H}}{\delta m} = -\partial_x^2 p_m + 2m(\partial_x p_m)^2 - 2(1 - \rho)\partial_x p_\rho \partial_x p_m - \text{Pe}(1 - \rho)\partial_x p_\rho + \sinh 2p_m.
\end{aligned}
$$

Their solutions inserted in the action Eq. (19) give $\Psi(\Lambda) = -\mathcal{S}_{\text{tot}}/(\ell_s T)$.

Note that in the limit of $T \gg 1$ the initial and final boundary conditions on the different fields do not play an important role, except for fixing the total particles mass $\rho_0$ (however, for an exception see [69]).

With the above formalism we now turn to solve the MFT equations. As a starting point we will consider a system in the homogeneous phase, i.e., away from MIPS. We comment about the extension to the MIPS state in the discussion Sec. 7.

## 4.1 Constant-profile solutions

The simplest solutions of the MFT equations obey the additivity principle [70] so that $\rho$ and $m$ are time independent, except for short time intervals around $t = 0$ and $t = T$, and the same holds for $\partial_x p_\rho$ and $\partial_x p_m$. The structure of the MFT equations implies that $p_m$ is time independent but that $p_\rho$ has a contribution growing linearly in time as $\nu t$ (see [71] for a similar problem). Assuming, in addition, translational invariance we find

$$\rho = \rho_0, \quad \nu = \Lambda, \quad m = \frac{\rho_0 C\Lambda}{\sqrt{1+(C\Lambda)^2}}, \quad p_m = \frac{1}{2}\operatorname{arcsinh}(C\Lambda), \tag{22}$$

with $C = \operatorname{Pe}(1-\rho_0)$. The resulting SCGF given by Eq. (17), that corresponds to the homogeneous solution $\Psi_H(\Lambda)$ is then

$$\Psi_H(\Lambda) = \rho_0(1-\rho_0)\Lambda^2 - \rho_0\left(1-\sqrt{1+C^2\Lambda^2}\right). \tag{23}$$

Performing the Legendre–Fenchel transform (16) we find

$$I_H(\rho_0, q) = q\Lambda - \Psi(\Lambda) = \rho_0(1-\rho_0)\Lambda^2 + \rho_0\left(1 - \frac{1}{\sqrt{1+C^2\Lambda^2}}\right), \tag{24}$$

with $\Lambda(\rho_0, q)$ given by the inverse of

$$q = \partial_\Lambda \psi(\Lambda) = 2\rho_0(1-\rho_0)\Lambda + \frac{\rho_0 C^2\Lambda}{\sqrt{1+C^2\Lambda^2}}. \tag{25}$$

Note that for $\operatorname{Pe} = 0$ we obtain

$$I_H(q, \rho_0, \operatorname{Pe} = 0) = \frac{q^2}{4\rho_0(1-\rho_0)}, \tag{26}$$

which coincides with the integrated current rate function of the simple symmetric exclusion process [45].[1] Indeed, in this case the self-propulsion of both types of particles is set to zero and the two models coincide.

As we now show, a non-zero self-propulsion ($\operatorname{Pe} \neq 0$) induces a dynamical phase transition. Indeed, the space-time constant solution (22) loses linear stability in a parameter regime that we identify. Linear instability can signal the onset of a *second* order dynamical phase transition, see, e.g. [60]. Nevertheless, as we find, the phase diagram here involves a *first* order transition.

As we now show, the full phase diagram, including first order transitions, can be derived by establishing an analogy with equilibrium phase separating systems with conserved order parameters.

## 5  Dynamical phase transitions

We now turn to show that the current large deviation function $I$ exhibits dynamical phase transitions as a function of $q, \rho_0, \operatorname{Pe}$. The analysis in this section focuses on the $\ell_s \to \infty$ asymptotics, where we recall that $\ell_s = \sqrt{\gamma/D_0}$ plays the role of the system length for the macroscopic coordinate $x \in [0, \ell_s]$. In the $\ell_s \to \infty$ limit, the system exhibits MIPS with sharp

---

[1]In fact, to make the comparison with the SSEP complete, one has to reinstate the variable rescaling $x \to x/\ell_s$ and $t \to t/\gamma$. This also results in a rescaling of the fluxes that enter in the un-scaled version of (6) and also enter in the definition (12). Taking all of these into account, one verifies the coincidence of the integrated current statistics of the two models.

domain walls whose width becomes independent of $\ell_s$. The finite-$\ell_s$ effects are studied in Sec. 6.

As we show, the signature of the DPT is the emergence of space- and time-dependent optimal profiles. To obtain these, one has to address the non-linear spatial and temporal minimization problem specified by Eqs. (21). As we now argue, the analysis in the $\ell_s \to \infty$ limit simplifies considerably. In this case, gradient terms in the action functional Eq. (18) can be neglected to leading order. This can be seen by noting that long-wavelength spatial modulations contribute as $O(\ell_s^{-1})$ to the action via gradient terms, which is much smaller than the $O(\ell_s)$ coming from the contribution of the bulk terms. Moreover, sharp domain walls with finite extension also have a subextensive $O(1)$ contribution to the action. This negligible cost of gradient terms in the $\ell_s \to \infty$ limit is reminiscent of the negligible interface tension terms in the free-energy functional of equilibrium phase-separating systems. In addition, as we show in Appendix C, optimal histories are either flat or given by sharply phase-separated profiles that travel at a constant velocity $V$. As such, the time derivative terms present in Eq. (18) can also be neglected; for the flat profile, the derivative terms are zero, and for the phase-separated profile, the contribution to the time derivative terms only arises from the localized boundaries which scale as $O(1)$ and can therefore be ignored.

All in all, the above implies that ideas very similar to those used in equilibrium phase separating systems can be used to analyze the phase diagram. In particular, optimal solutions can be found by minimizing a "free energy" functional for these fields, which is controlled by a bulk term given by the homogeneous rate function $I_H$ (24)

$$I \simeq \ell_s^{-1} \min_{\rho, J_\rho} \int_0^T dt \int_0^{\ell_s} dx \, I_H\big(\rho(x,t), J_\rho(x,t)\big), \tag{27}$$

subject to the constraints of the total mass conservation and the conditioning on the space-time averaged current (12)

$$\rho_0 = \frac{1}{\ell_s} \int_0^{\ell_s} dx \, \rho(x,t), \qquad q = \frac{1}{\ell_s} \int_0^{\ell_s} dx \, J_\rho(x,t). \tag{28}$$

Both $\rho$ and $J_\rho$ are spatially conserved by virtue of the integral constraints (28). The explicit derivation of the minimization problem (27) and (28) is presented in Appendix C. The terms omitted from the expression (27) are surface tension terms which, as explained above, have a sub-leading contribution.

In analogy with equilibrium phase separation, whenever the bulk rate function $I_H$ (24) becomes *locally* non-convex, the homogeneous solution (22) must lose linear stability. In the next section we derive the associated spinodal, that is shown to coincide with an explicit linear stability analysis of the full action (18) against spatial and temporal variations. Next, we turn to study the analogue of the binodal and the related first-order transition which corresponds to the loss of *global* convexity of $I_H$ (occurring when $I_H$ differs from its convex hull). The rate function $I$ (27) is then given by the convex hull of $I_H$.

## 5.1 Linear stability

The function $I_H(\rho_0, q)$ (24) is a convex function of $q$. Therefore, the linear instabilities can be identified by checking when the determinant of the Hessian becomes negative,

$$\frac{\partial^2 I_H}{\partial \rho_0^2} \frac{\partial^2 I_H}{\partial q^2} - \left( \frac{\partial^2 I_H}{\partial \rho_0 \partial q} \right)^2 < 0. \tag{29}$$

which can be expressed using the parametric representation, Eq. (24), and the differentiation rules

$$
\begin{aligned}
\frac{\partial}{\partial \rho_0}\bigg|_q &= \frac{\partial}{\partial \rho_0}\bigg|_\Lambda + \frac{\partial \Lambda}{\partial \rho_0}\bigg|_q \frac{\partial}{\partial \Lambda}\bigg|_{\rho_0} = \frac{\partial}{\partial \rho_0}\bigg|_\Lambda - \frac{\frac{\partial q}{\partial \rho_0}\big|_\Lambda}{\frac{\partial q}{\partial \Lambda}\big|_{\rho_0}} \frac{\partial}{\partial \Lambda}\bigg|_{\rho_0}, \\
\frac{\partial}{\partial q}\bigg|_{\rho_0} &= \frac{\partial \Lambda}{\partial q}\bigg|_{\rho_0} \frac{\partial}{\partial \Lambda}\bigg|_{\rho_0} = \frac{1}{\frac{\partial q}{\partial \Lambda}\big|_{\rho_0}} \frac{\partial}{\partial \Lambda}\bigg|_{\rho_0}.
\end{aligned}
\tag{30}
$$

Employing these, we find that the region where the Hessian determinant vanishes is given by the solution of $h_0 = 0$ with

$$
\begin{aligned}
h_0 &= \Lambda^2 \big\{ BC^2 \mathrm{Pe}(4C - B\,\mathrm{Pe}) \\
&+ 2A(1+C^2\Lambda^2)\big[4(1+C^2\Lambda^2)^2 + \sqrt{1+C^2\Lambda^2}(4C\,\mathrm{Pe} - B\,\mathrm{Pe}^2 + 4C^3\,\mathrm{Pe}\,\Lambda^2)\big]\big\} \\
&+ \Lambda^2 4BC^2\big[C^3\Lambda^2\,\mathrm{Pe} + (1+C^2\Lambda^2)^{3/2}\big],
\end{aligned}
\tag{31}
$$

where $\Lambda(q,\rho_0,\mathrm{Pe})$ is given by Eq. (25), $A = 2\rho_0(1-\rho_0)$, $B = 2\rho_0$, and $C = \mathrm{Pe}(1-\rho_0)$. A non-trivial solution to (31), with $\Lambda \neq 0$, emerges only for $\mathrm{Pe} > \mathrm{Pe}^* = \sqrt{2}$. The region of linear-instability grows with increasing Pe spreading from the point $(\rho_0 = 1, q = 0)$ of the phase space, as shown in Figs. 3 and 4. The saddle-point solutions are linearly unstable in the purple region.

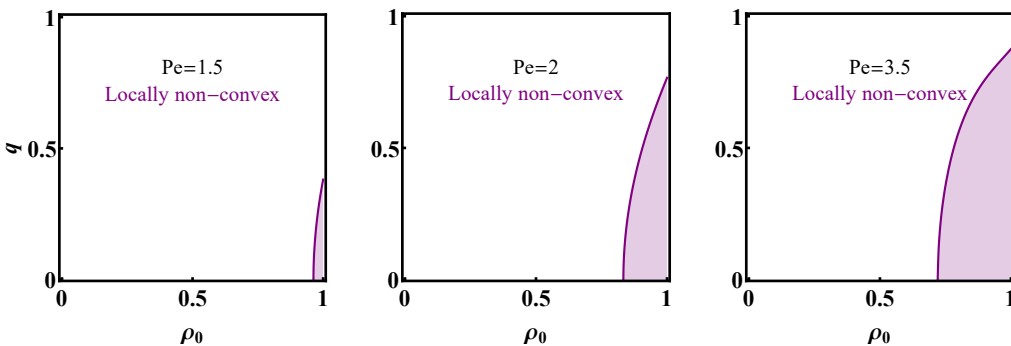

Figure 3: The emergence of local non-convexity of $I_H$ when increasing $\mathrm{Pe} > \mathrm{Pe}^* = \sqrt{2}$. The boundary of this region, marked by a purple line is given by the implicit relation $h_0 = 0$ with $h_0$ given in (31) together with (25).

A complementary derivation of this result is obtained by computing the second variation of the action functional Eq. (19) with respect to space and time dependent field variations at finite $\ell_s$ and then taking the limit $\ell_s \to \infty$. This calculation is given in Appendix D where we show explicitly that Eq. (31) is recovered (thus fully justifying the equilibrium-like analysis we put forward in this Section). The methodology used also allows us to explore the dynamical phase transitions for finite $\ell_s$, see Sec. 6.

As in equilibrium phase separation, the linear stability analysis alone does not allow one to obtain the optimal profile. The latter is determined by a global stability analysis. As we show below this allows us to identify a binodal for the transition. As in equilibrium, the linear stability analysis then identifies the spinodal which plays a role in determining the transient nucleation towards the final binodal decomposition, see e.g. [61] for related analysis. The study of such phenomenon is beyond the scope of the current study and would be an interesting subject for future investigations.

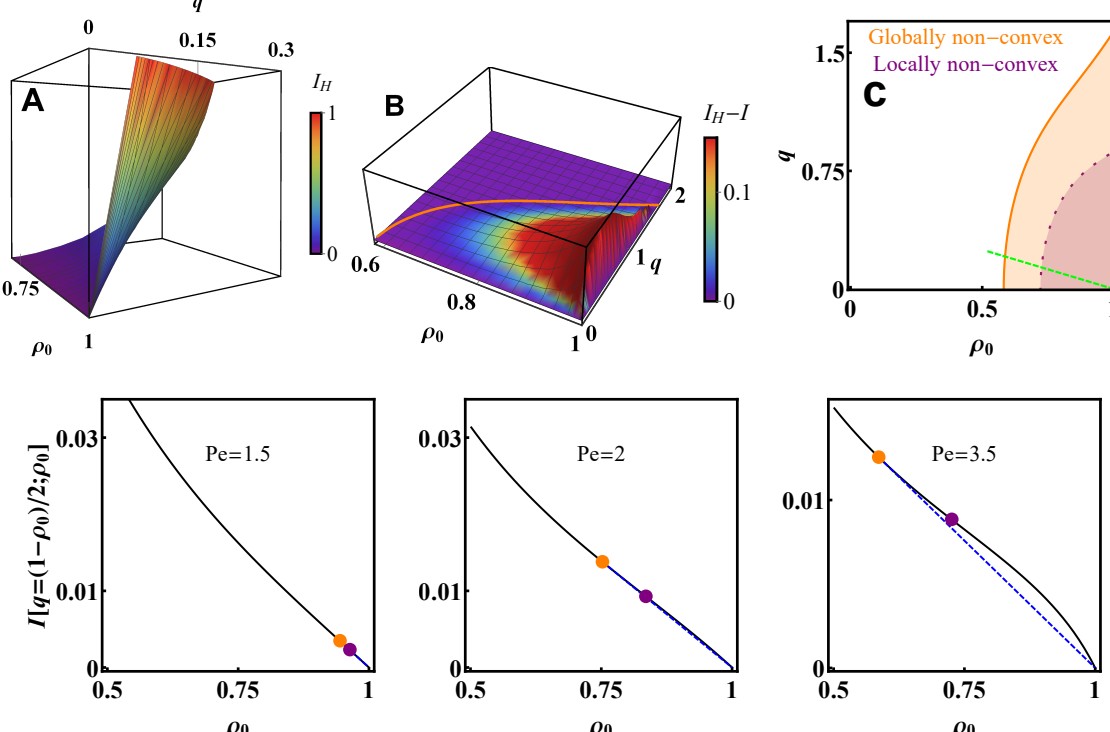

Figure 4: (A) The rate function $I_H(\rho_0, q)$ showing non-convexity and a pinch singularity at $(\rho_0 = 1, q = 0)$. (B) The difference between the rate function $I_H$ and its convex hull $I$. The orange line is the boundary of the region of global non-convexity, given by (33). (C) The phase diagram for local and global non-convexity for $I_H$. Local non-convexity is denoted by a dashed purple curve, given by the implicit relation $h_0 = 0$ with $h_0$ given in (31) together with (25). Global non-convexity is marked by the orange curve and is given by the implicit relation (33). At large $\ell_s$ these regions define local and global instability regions of the constant solution respectively (22). (Lower panels) Emergence of non convexity in $I_H$ at increasing Pe $> \sqrt{2}$. An example of a cross section for $I_H$, marked by a green line in panel (C) and given by $q = (1 - \rho_0)/2$. The dashed blue line is the convex hull construction. The orange and purple points corresponds to the intersection with the local and global convexity curves that are shown in panel (C).

## 5.2 Global stability

In Fig. 4 (A) and (B) we plot $I_H$ and the difference between $I_H$ and its convex hull as a function of $q$ and $\rho_0$ for Pe $= 3.5$. One can identify a curved surface with a concave region, which is reminiscent of a first-order phase transition in equilibrium systems. In the large-$\ell_s$ limit in which the domain wall contributions are negligible, one can then obtain the rate function by constructing the convex hull of $I_H$.

Interestingly, numerically we find that the convex hull always consists of tie lines between $(\rho_0 = 1, q = 0)$ and other points on the $I_H$ surface, e.g., the points on the orange line in Fig. 4(C). This is supported by the fact that $I_H$ has a singularity at $(\rho_0 = 1, q = 0)$. Indeed, $I_H$ presents global minima along the entire line $q = 0$, and diverges as approaching $\rho_0 = 1$. This results in a pinch-point singularity $I_H \sim q^2/(1 - \rho_0)$. The convex hull then needs to be constructed by drawing tie lines that pass through $(\rho_0 = 1, q = 0)$ and extend beyond the linearly unstable region. Denoting the density and the current in the low and high density phases by $(\rho_l, q_l)$ and $(\rho_h, q_h)$ respectively, then when coexistence occurs, the high density phase always

satisfies $\rho_h = 1$ and $q_h = 0$.

With this in mind, we are left with finding the location of the curve $(\rho_l, q_l)$ in the $(\rho_0, q)$ plane which defines the location of the low density phase. As explained above, it is found by demanding that a tie line emanating from the point $(\rho_h = 1, q_h = 0)$ is tangent to $I_H$ at the low density point $(\rho_l, q_l)$:

$$\left[ I_H + (1-\rho) \frac{\partial I_H}{\partial \rho} - q \frac{\partial I_H}{\partial q} \right]_{(\rho_l, q_l)} = 0 \,. \tag{32}$$

Using the differentiation rules (30) one then arrives at the relation:

$$h_1(\Lambda, \rho_0; \mathrm{Pe}) \equiv 1 - \sqrt{1 + C^2 \Lambda^2} + \Lambda^2 (1 - \rho_0)^2 (\sqrt{1 + C^2 \Lambda^2} + \mathrm{Pe}\, C) = 0 \,, \tag{33}$$

which together with the relation (25) defines the "binodal" curve for dynamical phase separation in an implicit form. The result for $\mathrm{Pe} = 3.5$ is shown in Fig. 4(C). Interestingly, the result implies that at large $\ell_s$ this transition starts at a vanishingly small value of $q$. The sub-leading interface tension cost in (27) shifts the critical value away from zero. As we detail in Sec. 6, its scaling is given by $q = O(\ell_s^{-1/2})$.

Note that for a given average density $\rho_0$, and total current $q$, the optimal configuration is phase-separated with the density and the current in each phase given by the geometric construction in the $(\rho_0, q)$ plane that is shown in Fig. 5. See also the lower panels of Fig. 4. The point $(\rho_l, q_l)$ is found by identifying a straight line between $(\rho_h = 1, q_h = 0)$ and the binodal line (33) which passes through the point of interest $(\rho_0, q)$. This implies, for example, that for a given density $\rho_0$, upon increasing the value of the current $q$ towards $q_l(\rho_0)$, the portion of the low density phase $\rho_l$ becomes larger, until it spans the entire system at $q = q_l(\rho_0)$.

Finally, we now show that when coexistence occurs, it is in the form of a traveling wave. To see this, we consider the interface between the high density phase with $(\rho_0 = 1, q = 0)$ and the low density one with $(\rho_l, q_l)$. A balance of fluxes implies that the domain wall between the two phases must propagate with a velocity

$$V = -\frac{J_l}{1 - \rho_l} = -\frac{q}{1 - \rho_0} \,, \tag{34}$$

where $J_l$ is the current in the low density phase. The second equality arises using $J_l f_l = q$ with $f_l$ the fraction of the low density phase so that $f_l \rho_l + (1 - f_l) = \rho_0$. A snapshot of this dynamics is illustrated in Fig. 5(B): A band of a high-density phase $\rho_h = 1$ propagates through the system in a direction opposite to the current in the low-density phase, as seen from the expression (34) of its velocity.

This sums up our findings for the rate function (13). For points in parameter space that are inside the homogeneous phase, see Fig. 5, it is given by $I = I_H$, with the homogeneous rate function $I_H$ given by (24). For points inside the traveling band phase it is given by

$$I = \frac{1 - \rho_0}{1 - \rho_l} I_H(\rho_l, q_l) \,, \tag{35}$$

with $\rho_l(\rho_0, q), q_l(\rho_0, q)$ given by the geometric construction involving the "binodal" (33). It has a jump in its second derivative at the point of intersection with the curve given by Eq. (33). These results are shown in Fig. 6.
Notice that finite-$\ell_s$ corrections leave the immediate neighborhood of $q = 0$ protected from the dynamical phase transition. These corrections will be discussed in detail in the next section.

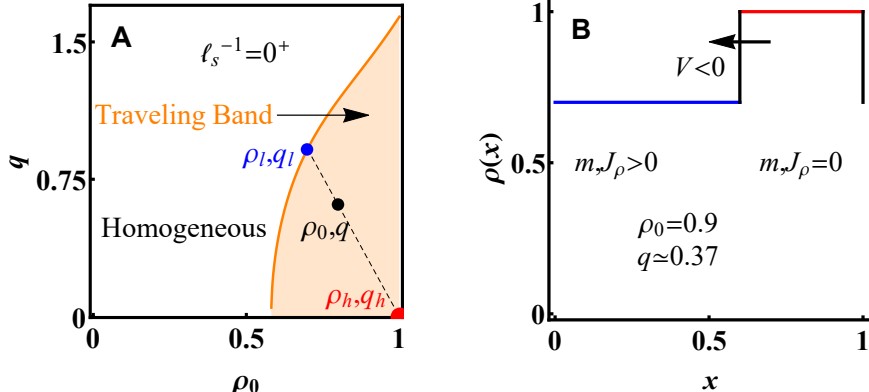

Figure 5: (A) The phase diagram for current fluctuations $q$ at Pe $= 3.5$. The orange curve marks the boundary between the traveling band phase and a homogeneous phase at $\ell_s \to \infty$. It is given in a parametric form by Eqs. (33) and (25). The dashed line denotes the convex hull construction between the high density phase which is always at close packing ($\rho_0 = 1, q = 0$), and the low density phase. It is given by the intersection of this line with the orange curve denoting the edge of the traveling band phase. (B) An instance of the traveling band solution that corresponds to the geometrical construction illustrated in panel (A). As discussed in Sec. 6, the sharp interface is smoothed at finite $\ell_s$.

# 6 Finite-$\ell_s$ corrections and relation to the MIPS criticality

Since both MIPS, which occurs at $q = 0$, and the travelling wave phase that we studied above exhibit phase separation, it is interesting to compare their phase diagrams and see how the transitions relate to each other. As we now show, this requires one to go beyond the analysis considered above and study the sub-leading finite-$\ell_s$ corrections. We also comment on the values of the cumulants in this limit at the end of the Section.

## 6.1 A primer: comparing MIPS and the DPT in the limit $\ell_s^{-1} \to 0$

Fig. 7 shows the phase diagrams of the two transitions in the limit of $\ell_s^{-1} \to 0$. The binodal for the DPT is plotted in orange in the $q \to 0$ limit, obtained from Eq. (33) using a small-$\Lambda$ expansion. This leads to the expression

$$\rho_{l,1} = \frac{1}{2} + \frac{1}{\text{Pe}^2} + O(q^2),\tag{36}$$

for the low density phase while the high density phase has a density $\rho_h = 1$. A similar small-$\Lambda$ expansion of Eq. (31) shows that the spinodal at $q \to 0$, plotted as a purple dashed line on Fig. 7, is given by

$$\rho_{l,2} = \frac{2}{3}\left(1 + \frac{1}{\text{Pe}^2}\right) + O(q^2).\tag{37}$$

A naive comparison with the phase diagram of MIPS (whose spinodal and binodal are shown as dashed and solid black lines respectively in Fig. 7) seems to suggest that the two transitions are not related. As we now show, this is in fact not the case. The link between MIPS and the DPT is revealed by studying finite-$\ell_s$ corrections to the rate function.

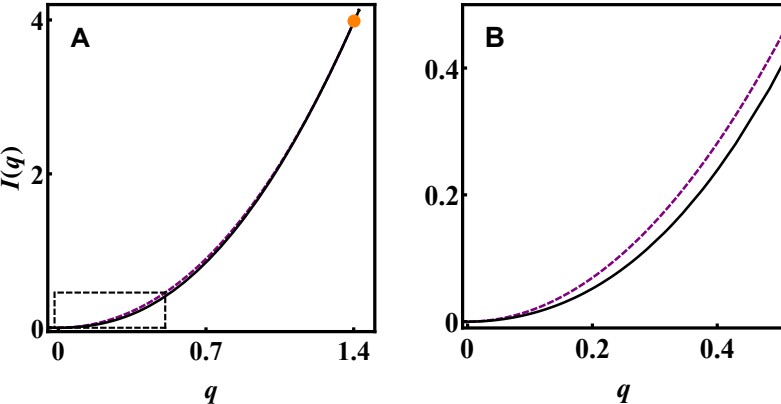

Figure 6: The rate function $I(q)$ of Eq. (13) at $\rho_0 = 0.9$ and $\ell_s^{-1} \to 0$. Panel (B) is a closeup of the region marked in panel (A) by a dashed rectangle. The continuous black line represents the rate function $I(q)$ while the dashed purple curve is the sub-optimal rate function that corresponds to a homogeneous profile (24). $I$ is constructed from the convex hull of $I_H$ in the parameter space $(\rho_0, q)$, see main text. The rate function has a jump in its second derivative at the orange point where these two rate functions are tangent. This point corresponds to the boundary of the traveling band phase denoted by the orange curve in Fig. 5 (A).

## 6.2 Relating the DPT at small $q$ to the MIPS at $q = 0$ through finite-$\ell_s$ corrections

We start by summarizing the results before turning to their derivation. To do this, we consider finite-$\ell_s$ corrections in the different regions of the phase diagram Fig. 7.

First, we note that outside the binodal (denoted by an orange line) there is no transition. Thus, as $q$ is increased from 0, the system remains homogeneous and there are no finite-$\ell_s$ corrections. Inside the orange binodal in the $\ell_s^{-1} \to 0$ limit, the system transitions to a sharply separated traveling wave at a finite $q$ (with two bulk densities dictated by the binodal). Accounting for a finite $\ell_s$ shows that such a state emerges only when $q$ becomes of the order $q_s \sim \ell_s^{-1/2}$. This is a result of the finite cost of the domain walls in the system. In other words, at finite $\ell_s$, there is a boundary layer $q \sim \ell_s^{-1/2}$ separating the $q = 0$ behavior (which display MIPS) from the DPT. In fact, by considering more closely the boundary layer we find two regions:

*Region I* – inside the spinodal ($\rho_{l,2} < \rho_0 < 1$ and $\mathrm{Pe} > \sqrt{2}$) [which encloses the MIPS critical point ($\rho_0 = 3/4, \mathrm{Pe} = 4$)].
In this region in the $\ell_s^{-1} \to 0$ limit the homogeneous state is *linearly* unstable for any $q > 0$. This is not longer true for large but finite $\ell_s$, where we find that the linear instability occurs at a finite threshold $q_{c,2}$, that scales as $q_{c,2} \sim \ell_s^{-1}$. When $q > q_{c,2}$, the systems exhibits a smooth spatial modulation which becomes more pronounced until a sharply-phase-separated state emerges at $q \sim \ell_s^{-1/2}$ (and the bulk densities of the profile are given to leading order by the orange binodal). Hence, the system first undergoes a linear instability into a smoothly modulated state and then crosses over to a sharply separated state over a range which scales as $O(\ell^{-1/2})$.

*Region II* – between the binodal and the spinodal ($\rho_{l,1} < \rho_0 < \rho_{l,2}$ and $\mathrm{Pe} > \sqrt{2}$). Here the system is linearly stable for any value of $\ell_s$. Then one has a discontinuous transition at $q_{c,1} \sim \ell_s^{-1}$ into a traveling wave solution whose domain walls become sharper with increasing $q$ as in region I.

Interestingly, at the critical point of the MIPS transition ($\rho_0 = 3/4, \mathrm{Pe} = 4$) both the DPT and the MIPS are initiated by a linear instability. This implies that the critical current $q_{c,2}$ of

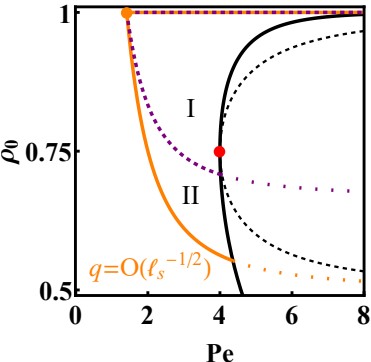

Figure 7: The phase diagrams of the DPT and of MIPS. The MIPS binodal is shown as a black solid line and the corresponding spinodal as a dashed black line. The binodal for dynamical phase transition at $q = O(\ell_s^{-1/2})$, which is given by Eq. (36), is plotted in orange. The dashed purple line is the corresponding spinodal, Eq. (37). Their extension in the MIPS phase is denoted by dotted lines.

the DPT vanishes at this point.

In sum, the MIPS transition, and the dynamical phase transition are separated by a finite layer at $q > 0$ everywhere in phase space, except at the MIPS critical point where the two transitions merge. This is presented in Fig. 8 and Fig. 9. We now provide a derivation of these results.

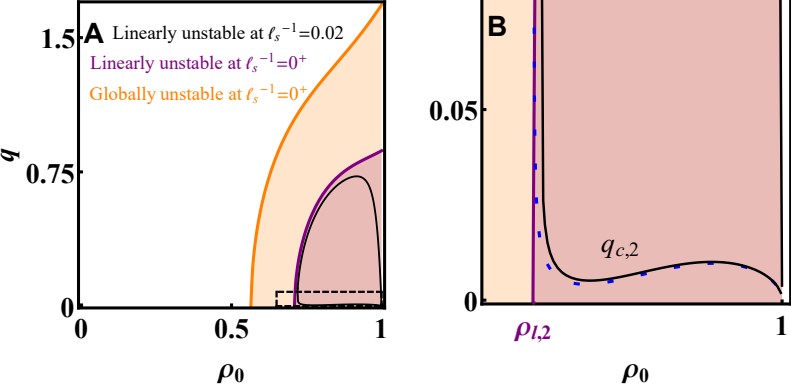

Figure 8: A comparison of the phase diagrams of the DPT in the infinite system-size limit ($\ell_s^{-1} = 0$) and at finite $\ell_s$, at Pe = 3.9. The region marked by a dashed box in panel (A) is shown in panel (B). As in Fig. 4 (C), the regions of linear and global instability for $\ell_s^{-1} \to 0$ are shown in purple and orange. The numerically obtained region of linear instability for a small but finite value $\ell_s^{-1} = 0.02$ is delimited by a black curve. The dashed blue line in panel (B) is the analytical expression (38) which is valid at first order for small $\ell_s^{-1}$, and is in good agreement with the previous numerical results. These curves illustrate the boundary layer described in the main text.

### 6.2.1 The scalings of $q_s$ and $q_{c,1}$

To evaluate the scalings of these current values, we study the effect of the sub-leading surface tension terms that were omitted in Eq. (27). Consider first a sharply phase separated state. Then, as explained at the beginning of Sec. 5, the contribution of interface terms to the rate

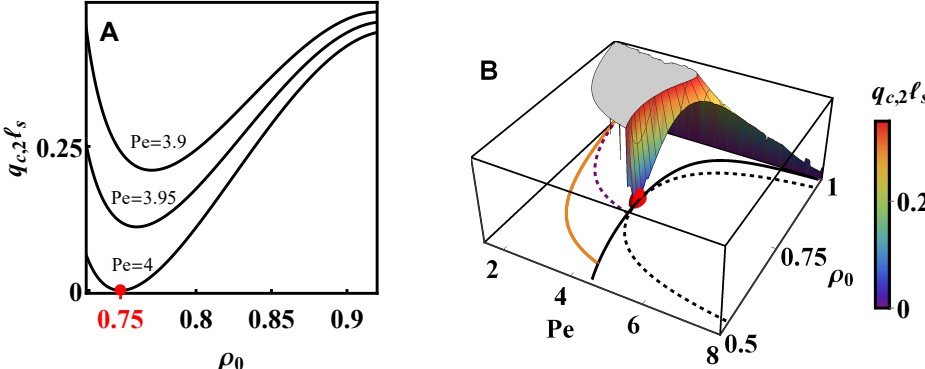

Figure 9: The critical current $q_{c,2}$, given by Eq. (38), for three Pe values. Panel (A) shows how the critical current (38) vanish at the MIPS critical density $\rho_0 = 3/4$ upon approaching the MIPS critical point Pe $= 4$. (B) The same as (A) but presented in the $(\rho_0, \text{Pe}, q)$ space. The critical current threshold, Eq. (38), vanishes at the MIPS critical point which is marked by the red dot. The black and dashed black lines are the MIPS binodal and spinodal respectively. The orange and purple curves are the boundaries of the globally and linearly unstable regions at $\ell_s^{-1} \to 0$ which are given by Eqs. (36) and (37), respectively.

function (27), evaluated over a sharply phase separated solution, scales as $O(\ell_s^{-1})$. Comparing this to the gain in the bulk term in the expression (27), which scale as $q^2$ for small $q$, we find that a sharply separated state only emerges when $q$ becomes of the order of $q_s \sim \ell_s^{-1/2}$.

Next, consider the transition into a smoothly modulated traveling wave state with a finite amplitude. In this case, the interface cost scales as $O(\ell_s^{-2})$. Correspondingly, the threshold value scales as $q_{c,1} \sim \ell_s^{-1}$. In the large $\ell_s$ limit, we therefore expect that the transition into a smoothly modulated profiles precedes the emergence of sharply phase separated traveling wave solutions everywhere in phase space.

### 6.2.2 Critical current $q_{c,2}$ in region I

The finite critical value $q_{c,2}$ can be obtained exactly for any $\ell_s$, from a stability analysis of the action (19) to small space-and-time dependent fluctuations. As detailed in Appendix E, one finds

$$q_{c,2} = \pi \ell_s^{-1} \frac{\sqrt{(2 + CD)^2 (A + \rho_0 C^2)}}{\sqrt{\text{Pe}^2 (3\rho_0 - 2) - 2}} + O(\ell_s^{-2}), \tag{38}$$

with $A(\rho_0) = 2\rho_0(1 - \rho_0)$, $C(\rho_0, \text{Pe}) = \text{Pe}(1 - \rho_0)$ and $D(\rho_0, \text{Pe}) = \text{Pe}(1 - 2\rho_0)$. The expression under the square root in the denominator is positive in the linearly unstable region $\rho_0 > \rho_{l,2}$, see Eq. (37). Importantly, the scaling $q_{c,2} \sim \ell_s^{-1}$ guarantees that linear instability occurs *before* the transition into the sharply separated state in this region (which occurs at $q \sim \ell_s^{-1/2}$).

These results are presented in Fig. 8 and Fig. 9. Notice that, as argued above, $q_{c,2}$ vanishes at the MIPS critical point ($\rho_0 = 2/3$, Pe $= 4$), given by $2 + CD = 0$. Indeed, from Eq. (38), $q_{c,2}$ is proportional to $|2 + CD|$; however, $q_{c,2} > 0$ everywhere inside the region enclosed by the MIPS spinodal $2 + CD < 0$. The reason is that the linear instability into MIPS (at $q = 0$) is distinct from the transition into a traveling wave (at $q > 0$). They only coincide at the MIPS critical point. Interestingly, the distinction between the transitions can be seen in the spectrum of the linearized problem. The MIPS linear instability describes an exponential growth of spatial modulations towards a state which is phase separated and *stationary*. Therefore, the corresponding excitation frequency $\omega$ that describe this linear instability is *imaginary*. In

contrast, the linear instability associated with the dynamical phase transition is characterized by the emergence of traveling waves. Therefore, its associated excitation frequency $\omega$ is *real*.

It is only at the MIPS critical point that the excitation frequencies of both instabilities vanish and the two transitions merge.

### 6.3  Remarks on the current cumulants and finite-$\ell_s$ effects

As usual in large-deviation theory, the rate function $I(q)$ is related to the cumulants of the current through derivatives of $I$ at $q = 0$ [46]. Notice that, in the $\ell_s \to \infty$ limit, the curvature at $q = 0$ of the rate function $I$ differs from that of $I_H$, the rate function for a homogeneous profile (shown in Fig. 6), since $I$ is the convex envelope of $I_H$. However, at large but finite $\ell_s$, as we have seen in the previous paragraph, the rate function exhibits a small boundary layer around $q = 0$, in which it is equal to $I_H$. This means that the cumulants of the current are in fact obtained from the set of derivatives of $I_H(q)$ at $q = 0$. The presence of a DPT in the rate function of the current is in some cases associated to an anomalous scaling of the finite-size corrections to the rate function (see e.g. [57] for the cumulants of the current in the weakly asymmetric simple exclusion process (WASEP)). This is an interesting open question which remains to be addressed for this problem.

In fact we can draw a parallel between the large deviations of the current in our problem and in the (asymmetric) exclusion processes: in the periodic WASEP, a DPT for the current large deviation is observed between a homogeneous phase and an heterogeneous-profile phase, with smooth interfaces [55]. As the asymmetry is increased, the interfaces become sharper and a DPT point occurs closer to the average current, making the current distribution more singular. In the very large asymmetry limit, one recovers the TASEP (totally asymmetric simple exclusion process). When doing so, the dynamical scaling switches from diffusive to KPZ [54] and the nature of the fluctuations change considerably. In our case, in the presence of phase separation, the domains are separated by sharp walls only in the large-$\ell_s$ limit, and the location of the DPT becomes closer to the average current which occurs at $q = 0$ in the same limit – the situation therefore is quite similar, except that the dynamical scaling remains diffusive all along the limiting process. This is also what allows one to safely determine the current cumulants in this limit, although the system presents sharp walls.

## 7  Summary and discussion

In this work we have derived the exact fluctuating hydrodynamics for the active lattice gas model of Ref. [39]. This is the first such derivation for a system of interacting active particles. We note here that the active lattice gas model differs from standard active matter models. In contrast to more standard models, the tumbling rate is rescaled by $L^{-2}$, see Sec. 2. As a result, the polarization enters as a slow field in the coarse-grained hydrodynamics. Still, this model shares many similarities with the more standard active matter models, most notably MIPS.

Building on the fluctuating hydrodynamics, we have extended the classical MFT framework to the active lattice gas model and employed it to study the integrated current fluctuations. We provide a full mapping of the current fluctuations phase diagram where, notably, we identify a dynamical phase transition. The MFT problem could be tackled analytically despite the fact that it is significantly more involved compared to the standard Gaussian noise case. Here we find Poissonian noise and the hydrodynamics involve two scalar fields rather than one. Nevertheless, the analysis can be carried out analytically using $\ell_s^{-1}$ as a small parameter.

For the unbiased system, this small parameter controls the ratio of the domain wall width in the MIPS phase to the system size. As we have shown, the same small parameter sets a vanishing probability cost for gradient terms in the MFT action. This enables us to establish an

analogy between the MFT action and the free energy of equilibrium phase separating systems. A non-convexity of the bulk term of the 'free energy' can then be associated with a dynamical phase transition into a traveling wave phase separated state. We find that this transition occurs at vanishingly small $O(\ell_s^{-1})$ current fluctuations. Interestingly, the dynamical phase transition of the biased system and the MIPS of the unbiased system are shown to coincide at the MIPS critical point where both of these are initiated through a linear instability. This link is exposed by accounting for finite-$\ell_s$ corrections to the MFT problem.

Our results were derived assuming that the unbiased system is homogeneous. Nevertheless, we anticipate that to leading order at small $\ell_s^{-1}$, the results will hold also in the MIPS phase. In particular, the minimization determined by the procedure described above which allows for the biased system to be phase separated.

It is interesting to use the formalism developed above to study other large-deviation functions. In particular, there has been recent interest in entropy production rates in active systems (even at the absence of any external drive either at the bulk or at the boundaries). The study of entropy production for this active lattice gas is the subject of a future publication [72].

From a broader perspective, it could be interesting to investigate other models presenting a diffusive scaling but non-Gaussian fluctuations coming from processes other than the specific case of tumbling events we have studied here (for instance from chemical reactions or non-Gaussian sources of noise).

## Acknowledgements

TA is funded by Blavatnik Postdoctoral Fellowship Program. VL is supported by the ANR-18-CE30-0028- 01 Grant LABS and by the EverEvol CNRS MITI project. SR and YK are supported by an Israel Science Foundation grant (2038/21) and an NSF-BSF grant (2016624).

## A  Deriving the large deviation function

### A.1  Contributions from conserved fluxes

Flux fluctuations are given by the Gaussian noise terms $J_\rho - \bar{J}_\rho = \eta_\rho / \sqrt{L\ell_s^{-1}}$ and $J_m - \bar{J}_m = \eta_m / \sqrt{L\ell_s^{-1}}$[2] with the covariances

$$
\begin{aligned}
\langle \eta_\rho(x,t)\eta_\rho(x',t') \rangle &= \sigma_\rho \, \delta(x-x')\delta(t-t'), \\
\langle \eta_m(x,t)\eta_m(x',t') \rangle &= \sigma_m \, \delta(x-x')\delta(t-t'), \\
\langle \eta_\rho(x,t)\eta_m(x',t') \rangle &= \sigma_{\rho,m} \, \delta(x-x')\delta(t-t'),
\end{aligned} \tag{A.1}
$$

given by

$$
\sigma_\rho = 2\rho(1-\rho), \quad \sigma_m = 2(\rho - m^2), \quad \sigma_{\rho,m} = 2m(1-\rho).
$$

These have been derived in Ref. [43] using a mapping to the ABC model [64–66]. Importantly, as shown in Ref. [66], these also capture non-typical fluctuations of $O(1)$.

---

[2]Notice the factor $\ell_s^{-1}$ in the noise amplitude, as compared to Ref. [43]. It follows from the rescaling of space and time that we employ throughout this work, see the definition bellow Eq. (1).

## A.2 Contributions from tumbling events

The Gaussian approximation to the tumbling probability was derived in Ref. [43]. Here we go beyond this approximation by accounting for the underlying Poisson process.

To account for the probability of tumbles we note that these are transmutation reactions between two particle species. Such processes were treated in Ref. [62, 63, 68]. One finds that the probability $P(\mathcal{K}_+)$ to observe $\mathcal{K}_+$ flips of $+$ particles into $-$ particles in a space interval $\Delta x$ and time interval $\Delta t$ follows a large deviation principle

$$-\ln P\left(\mathcal{K}_+ = K_+ L \ell_s^{-1} \Delta t \Delta\right) \simeq L \ell_s^{-1} \Delta x \Delta t \psi_{\rho_+}(K_+), \tag{A.2}$$

with the expected large-deviation function of a Poisson process:

$$\psi_{\rho_+}\left(K_+\right) = K_+ \log\left(\frac{K_+}{\rho_+}\right) - K_+ + \rho_+. \tag{A.3}$$

Here $\rho_+$ is the density of $+$ particles in the mesoscopic interval $[x, x + \Delta x]$. Similarly, the probability $P(\mathcal{K}_-)$ to observe $\mathcal{K}_-$ flips of $-$ particles into $+$ particles in a space interval $\Delta x$ and time interval $\Delta t$ follows a similar large deviation principle with

$$\psi_{\rho_-}(K_-) = K_- \log\left(\frac{K_-}{\rho_-}\right) - K_- + \rho_-. \tag{A.4}$$

Here $\rho_-$ is the density of $+$ particles in the mesoscopic interval $[x, x + \Delta x]$.

The total change in the number of $-$ particles in a mesoscopic interval, due to tumbling reactions is then given by the difference $\mathcal{K} = \mathcal{K}_+ - \mathcal{K}_-$. Following the contraction principle [46], the probability of this variable is described by the large deviation function

$$-\ln P\left(\mathcal{K} = K L \ell_s^{-1} \Delta t \Delta x\right) \simeq L \ell_s^{-1} \Delta x \Delta t \mathcal{L}_K(K), \tag{A.5}$$

where $\mathcal{L}_K$ is found by minimizing the combined probability cost of the previous two processes under the constraint of a given total tumbling rate

$$\mathcal{L}_K = \inf_{K_+}\left[\psi_{\rho_+}(K_+) + \psi_{\rho_-}(K_+ - K)\right] = \rho - \sqrt{K^2 + (\rho^2 - m^2)} + K \ln\left[\frac{\sqrt{K^2 + (\rho^2 - m^2)} + K}{(\rho + m)}\right]. \tag{A.6}$$

## A.3 Joint large deviation function

The above results can be combined into a single large deviation function as the conserved dynamics of the fluxes are uncorrelated with the tumbling dynamics. The microscopic rates for tumble events are much slower than the hopping dynamics. In particular, this allows one, despite the presence of tumbling events, to use a local equilibrium conditions for the hopping rates. In sum, this means that the total action is written as a sum of the (Gaussian) flux action and the tumbling action $\mathcal{L}_K$, as announced in Eq. (8).

Notice that, as expected, the quadratic expansion $\mathcal{L}_K(K) \simeq (K - m)^2 / 2\rho$ close to the minimum the joint large-deviation function exactly matches the Gaussian noise term derived in Ref. [43].

# B Deriving the MFT equations

In this Appendix, we derive the action (18) starting from the path integral (17). To do so we use the Martin–Siggia–Rose–Janssen–de-Dominicis formalism [50, 73–75]. The generating

function then takes the form

$$e^{LT\Psi(\Lambda)} = \int \mathcal{D}\rho\,\mathcal{D}m\,\mathcal{D}J_\rho\,\mathcal{D}J_m\,\mathcal{D}K\,\mathcal{D}\hat{p}_\rho\,\mathcal{D}p_m \times$$

$$e^{-L\ell_s^{-1}\left\{\hat{\mathcal{S}}_{\mathcal{L}} - \int_0^{\ell_s} dx \int_0^T dt\left[\Lambda J_\rho - \hat{p}_\rho(\dot\rho + \partial_x J_\rho) - p_m(\dot m + \partial_x J_m + 2K)\right]\right\}} . \tag{B.1}$$

with two auxiliary response fields $\hat{p}_\rho(x,t)$ and $p_m(x,t)$ arising from representing the $\delta$ functions in Eq. (17) using a Fourier transform. To obtain the MFT equations we then use a saddle-point evaluation at large $L$. Minimizing with respect to $K$ one arrives at the optimal value

$$K = m\cosh 2p_m - \rho\sinh 2p_m . \tag{B.2}$$

Next, minimizing with respect to the flux fields $J_\rho$ and $J_m$ yields

$$\begin{bmatrix} J_\rho - \bar{J}_\rho \\ J_m - \bar{J}_{m\cdot} \end{bmatrix} = \mathbf{C}\begin{bmatrix} \partial_x\hat{p}_\rho + \Lambda \\ \partial_x p_m \end{bmatrix} . \tag{B.3}$$

Finally, using the expressions (B.2) and (B.3), one arrives at the announced action (18).

Notice that since the fields $\rho, m, J_\rho, J_m, K$ are continuous, they obey periodic boundary conditions on the ring geometry as do the fields $\hat{p}_\rho$ and $p_m$. As stated in Sec. 4, the boundary conditions *in time* for the optimal fields become irrelevant in the large-$T$ limit.

## C Establishing an equilibrium analogue

In this Appendix we show how the rate function $I(q)$ of Eq. (13) can be found, to leading order at large $\ell_s$, by the minimization problem (27) subjected to the constraints (28). It is more convenient to start from the Lagrangian formulation (8), rather then the Hamiltonian one (19). From Eqs. (8) and (13), we have

$$I(q) = \frac{1}{\ell_s T}\min_{\rho, m, J_\rho, J_m, K}\int_0^{\ell_s} dx \int_0^T dt\,(\mathcal{L}_J + \mathcal{L}_K) , \tag{C.1}$$

subject to the integrated current constraint (12), and the dynamical constraints (6). In contrast to the Hamiltonian formulation (19), these constrains are not built into the Lagrangian minimization and have to be enforced explicitly.

At large times, if the additivity principle is verified, optimal solutions are time independent and we have shown in Sec. 4.1 that this leads to homogeneous density and polarization optimal profiles, with a corresponding rate function $I_H$ given by Eq. (24). When the additivity principle is broken, it happens in general with optimal profiles taking the form of traveling waves that propagate at a constant velocity. This is the form that we assume now. Furthermore, in the large-$\ell_s$ asymptotics, the width of the walls is $O(1)$: it does not scale with the system size $\ell_s$. As discussed in the main text, the contribution of the domain walls to the action is then sub-extensive. We now explain how, for such profiles, the rate function $I(q)$ can still be obtained from the homogeneous one $I_H$ with the adequate constraints – in a picture analogous to what happens in equilibrium phase separation.

The minimization in (C.1) becomes time-independent for the optimal traveling profiles $\{\rho, m, J_\rho, J_m, K\}$ (which depend only on $x$):

$$I(q) = \frac{1}{\ell_s}\min_{\{\rho, m, J_\rho, J_m, K\}}\int_0^{\ell_s} dx\,(\mathcal{L}_J + \mathcal{L}_K) . \tag{C.2}$$

The time dependency is also eliminated from the current constraint (12)

$$q = \frac{1}{\ell_s} \int_0^{\ell_s} dx \, J_\rho(x). \tag{C.3}$$

We next consider the dynamical constraints (6). The first one implies that the total mass is conserved, which we now assume implicitly:

$$\rho_0 = \frac{1}{\ell_s} \int_0^{\ell_s} dx \, \rho(x). \tag{C.4}$$

Also, it implies that the traveling wave solution moves at a constant speed

$$V = \frac{J_h - J_l}{\rho_h - \rho_l}, \tag{C.5}$$

where $J_{h,l}$ and $\rho_{h,l}$ are the high and low density values for the flux and total density. The second constraint in Eq. (6), $\partial_t m = -\partial_x J_m - 2K$, implies that optimal tumbling rate vanishes in the bulk phases

$$K = 0. \tag{C.6}$$

We are thus left with the four field minimization

$$I(q) = \frac{1}{\ell_s} \min_{\{\rho, m, J_\rho, J_m\}} \int_0^{\ell_s} dx \, \mathcal{L}, \tag{C.7}$$

$$\mathcal{L} = \left\{ \frac{1}{2} \begin{bmatrix} J_\rho - \mathrm{Pe} \, m(1-\rho) \\ J_m - \mathrm{Pe} \, \rho(1-\rho) \end{bmatrix}^{\mathrm{T}} \mathbf{C}^{-1} \begin{bmatrix} J_\rho - \mathrm{Pe} \, m(1-\rho) \\ J_m - \mathrm{Pe} \, \rho(1-\rho) \end{bmatrix} + \rho - \sqrt{\rho^2 - m^2} \right\}, \tag{C.8}$$

subject to the integrated current constraint (C.3), which can be imposed via a *scalar* Lagrange multiplier $\Lambda$.

In (C.8) we have omitted negligible gradient terms, but one has to be cautious because this implies that the Euler–Lagrange equations become algebraic and only admit constant value solutions. To retain possible non-homogeneous optimal solutions one introduces a space-dependent Lagrange multiplier $\tilde{\Lambda}(x)$ enforcing $J_\rho(x)$ to be equal to a profile $\tilde{J}_\rho(x)$. One arrives at

$$I(q) = \frac{1}{\ell_s} \min_{\{\rho, m, J_\rho, J_m, \tilde{\Lambda}, \tilde{J}_\rho, \Lambda\}} \left\{ \int_0^{\ell_s} dx \left[ \mathcal{L} + \tilde{\Lambda}(x) \big( \tilde{J}_\rho(x) - J_\rho(x) \big) + \Lambda \big( \tilde{J}_\rho(x) - q \big) \right] \right\}, \tag{C.9}$$

where again $\rho, m, J_\rho, J_m, \tilde{\Lambda}, \tilde{J}_\rho$ depend on $x$ while $\Lambda$ is constant. Omitting the dependencies on $x$ for simplicity and minimizing with respect to the flux fields we find

$$\begin{bmatrix} J_\rho - \mathrm{Pe} \, m(1-\rho) \\ J_m - \mathrm{Pe} \, \rho(1-\rho) \end{bmatrix} = \mathbf{C} \begin{bmatrix} \tilde{\Lambda} \\ 0 \end{bmatrix}. \tag{C.10}$$

Replacing these into (C.9) and minimizing with respect to $m$ then gives

$$m = \frac{\rho \, \mathrm{Pe}(1-\rho)\tilde{\Lambda}}{\sqrt{1 + (\mathrm{Pe}(1-\rho)\tilde{\Lambda})^2}}, \tag{C.11}$$

which is the same expression as was obtained for the homogeneous solutions in Eq. (22), but with $\Lambda$ replaced by space-dependent field $\tilde{\Lambda}$. Minimizing with respect to $\tilde{\Lambda}$ yields the equation

$$\tilde{J}_\rho = 2\rho(1-\rho)\tilde{\Lambda} + \frac{\rho \, \mathrm{Pe}^2(1-\rho)^2\tilde{\Lambda}}{\sqrt{1 + \mathrm{Pe}^2(1-\rho)^2\tilde{\Lambda}^2}}, \tag{C.12}$$

which is again, a space-dependent version of Eq. (25). Last, minimizing with respect to $\Lambda$ merely gives the constraint

$$q = \frac{1}{\ell_s} \int_0^{\ell_s} dx\, \tilde{J}_\rho \,. \tag{C.13}$$

Gathering the previous results into Eq. (C.8), we arrive at

$$I = \frac{1}{\ell_s} \min_{\rho(x),\tilde{J}_\rho(x)} \int_0^{\ell_s} dx \left\{ \rho(1-\rho)\tilde{\Lambda}^2 + \rho\left(1 - \left[1 + \text{Pe}^2(1-\rho)^2\tilde{\Lambda}^2\right]^{-1/2}\right) \right\}, \tag{C.14}$$

where $\tilde{\Lambda}$ is given implicitly by the algebraic relation (C.12), and $\tilde{J}_\rho$ satisfies (C.13). Comparing Eqs. (C.14) and (C.12) with (24) and (25), we conclude that

$$I = \frac{1}{\ell_s} \min_{\rho(x),\tilde{J}_\rho(x)} \int_0^{\ell_s} dx\, I_H\left[\rho(x),\tilde{J}_\rho(x)\right], \tag{C.15}$$

subject to the constraint (C.13) (and the total mass constraint (C.4) which was assumed all along). Changing the dummy variable $\tilde{J}_\rho(x)$ to $J_\rho(x)$ we finally arrive at Eqs. (27) and (28) which concludes our proof.

# D  Action second variation analyses and its relation to local concavity of $I_H$

We expand the action to second order in path variations around the homogeneous solutions (22). To do so, we define the variation fields $\delta\rho$, $\delta m$, $\delta p_\rho$, and $\delta p_m$ through[3]

$$\rho = \rho_0 + \delta\rho\,, \qquad\qquad p_\rho = \Lambda x + i\delta p_\rho\,,$$
$$m = \frac{\rho_0 C\Lambda}{\sqrt{1+(C\Lambda)^2}} + \delta m\,, \quad p_m = \frac{1}{2}\text{arcsinh}(C\Lambda) + i\delta p_m\,, \tag{D.1}$$

and the Fourier transforms

$$\delta f(x,t) = \sum_{n,m}, \delta f^{n,m} e^{i(k_n x + \omega_m t)}, \qquad k_n = 2\pi n/\ell_s, \qquad \omega_m = 2\pi m/T\,, \tag{D.2}$$

where $\delta X$ is the variation of the field $X$ and $n$ and $m$ are integers. After expansion, the quadratic variation term of the action Eq. (19), which we denote by $\delta S_2$, takes the form

$$\delta S_2 = \ell_s T \sum_{n,m} V_{-n,-m}^T \mathbf{B}(n,m) V_{n,m}\,, \tag{D.3}$$

with the matrix

$$\mathbf{B}(n,m;\Lambda,\rho_0,\text{Pe}) = \tag{D.4}$$

$$\begin{pmatrix} \Lambda^2 & \frac{\omega_m}{2} + \frac{Dk_n\Lambda}{\text{Pe}} - \frac{k_n BC\Lambda\text{Pe}}{4\sqrt{1+(C\Lambda)^2}} + i\frac{k_n^2}{2} & \frac{\text{Pe}\Lambda}{2} & \frac{k_n}{2}\left(D - \frac{BC\Lambda^2}{\sqrt{1+(C\Lambda)^2}}\right) - iC\Lambda \\ -\frac{\omega_m}{2} - \frac{Dk_n\Lambda}{\text{Pe}} + \frac{k_n BC\Lambda\text{Pe}}{4\sqrt{1+(C\Lambda)^2}} + i\frac{k_n^2}{2} & \frac{Ak_n^2}{2} & -\frac{k_n C}{2} & \frac{BC^2 k_n^2\Lambda}{2\text{Pe}\sqrt{1+(C\Lambda)^2}} \\ \frac{\text{Pe}\Lambda}{2} & \frac{k_n C}{2} & 0 & \frac{\omega_m}{2} + \frac{k_n C\Lambda}{\text{Pe}} + i\frac{k_n^2}{2} + i\sqrt{1+(C\Lambda)^2} \\ -\frac{k_n}{2}\left(D - \frac{BC\Lambda^2}{\sqrt{1+(C\Lambda)^2}}\right) - iC\Lambda & \frac{BC^2 k_n^2\Lambda}{2\text{Pe}\sqrt{1+(C\Lambda)^2}} & -\frac{\omega_m}{2} - \frac{k_n C\Lambda}{\text{Pe}} + i\frac{k_n^2}{2} + i\sqrt{1+(C\Lambda)^2} & k_n^2\left(\frac{B}{2} - \frac{B^2(C\Lambda)^2}{4(1+(C\Lambda)^2)}\right) + \frac{B}{\sqrt{1+(C\Lambda)^2}} \end{pmatrix},$$

---

[3]Note that the conjugate momentum variations are imaginary since the field is imaginary. It only takes a real value at the saddle point, see e.g. Ref. [61] and also Appendix B.

where $A = 2\rho_0(1-\rho_0)$, $B = 2\rho_0$, $C = \mathrm{Pe}(1-\rho_0)$, $D = \mathrm{Pe}(1-2\rho_0)$, and

$$V_{n,m} = \begin{bmatrix} \delta\rho^{n,m} \\ \delta p_\rho^{n,m} \\ \delta m^{n,m} \\ \delta p_m^{n,m} \end{bmatrix}. \tag{D.5}$$

The homogeneous solution becomes unstable when one of the eigenvalues of the matrix $\mathbf{B}$ becomes negative for some mode $n, m$. To find the region in parameter space $(\rho_0, \Lambda)$ where this happens we first note that by direct inspection of the eigenvalues of $\mathbf{B}$, and similar to related problems [60,61], the most unstable mode is $n = 1$. This is expect as the larger values of $n$ have a larger cost in the action due to spatial modulations.

For this mode we find that for any unstable point in the parameter space $(\rho_0, \Lambda)$, there exists a set of unstable frequencies $\omega_m$. In the large $T$ limit, the frequency can be treated as a continuous variable $\omega$ and the set of unstable frequencies become an interval. This interval shrinks to a point at the boundary of the unstable region. The boundary of the unstable region can then be identified by solving the equations

$$\mathrm{Det}[\mathbf{B}](\omega; n = 1, \Lambda, \rho_0, \mathrm{Pe}, \ell_s) = 0 \quad ; \quad \partial_\omega \mathrm{Det}[\mathbf{B}](\omega; n = 1, \Lambda, \rho_0, \mathrm{Pe}, \ell_s) = 0. \tag{D.6}$$

The first equation ensures that there are eigenvalues whose value is zero, while the second equation ensures that there is only one such eigenvalue.[4] The two equations (D.6) can be cast into a single implicit algebraic relation between $\Lambda$ and $\rho_0$ for given values of Pe and $\ell_s$ which we write as:

$$h_2(\Lambda, \rho_0; \mathrm{Pe}, \ell_s) = 0. \tag{D.7}$$

Using Eq. (25) this relation then defines a curve in the parameter space $(\rho_0, q)$ which encloses the linearly unstable region. This curve is plotted as a black line in Fig. 8 for $\ell_s^{-1} = 0.02$ and Pe = 3.9. The figure also shows, as a purple line, the limiting curve that is approached as $\ell_s \to \infty$.

As stated in the main text, the curve defined Eq. (D.7), which describes the local instability, must coincides with the curve Eq. (31), which describes local non-convexity. That is

$$\lim_{\ell_s \to \infty} h_2(\Lambda, \rho_0; \mathrm{Pe}, \ell_s) = h_0(\Lambda, \rho_0; \mathrm{Pe}). \tag{D.8}$$

This can be shown explicitly by expanding $h_2$ at large $\ell_s$, assuming a travelling wave form for the unstable mode, and taking the limit $\ell_s \to \infty$. Since the derivation is rather lengthy but straightforward, we omit it from the text.

# E   Deriving the critical threshold $q_{c,2}$ Eq. (38)

The curve $q_{c,2}$, given by Eq. (38) serves as a finite-$\ell_s$ correction to the limit (D.8). To account for it we must retain the next order expansion to the determinant keeping terms of $O(\ell_s^{-4})$. In addition, being interested only the region near $q = 0$, we use the scaling $\Lambda = O(\ell_s^{-1})$. This also implies the same scaling for the wave velocity $V = \omega k_1 = O(\ell_s^{-1})$. Using these and expanding up to $O(\ell_s^{-4})$ we now arrive at the explicit expression

$$\Lambda^2 = (2\pi)^2 \ell_s^{-2} \frac{(2 + CD)^2}{(2A + BC^2)(B\mathrm{Pe}^2 - 4 - 4\mathrm{Pe}C)}, \tag{E.1}$$

---

[4]We find numerically that for each point in parameter space there is only a single interval of frequencies where the determinant drops bellow zero.

with $A, B, C$ and $D$ defined in Eq. (D.4). Finally, expanding Eq. (25), we find

$$\Lambda(q) = \frac{2q}{2A + BC^2} + O(q^3), \tag{E.2}$$

which, used into Eq. (E.1), yields Eq. (38).

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
