# Peer review of "Macroscopic Fluctuation Theory and Current Fluctuations in Active Lattice Gases"

_SciPost Physics, doi:SciPost Phys. 14, 045 (2023)_

## Round 1 · Referee Report · Anonymous · 2022-10-23

Report

In this paper, the authors study a discrete-space and continuous time model
of active particles, which was introduced in Ref. [39]. At variance with standard
active particle systems, the tumbling rate in this model is scaled with the system size
in such a way that all the processes occur on diffusive time scales. Despite of this, the
model shares many similarities with more standard active matter models, in particular
motility induced phase separation (MIPS). One of the main results of the paper is to derive
exact fluctuating hydrodynamic equations (in the limit of a large system).

In a previous work [43], the same authors, using Macroscopic Fluctuation Theory (MFT),
studied the typical fluctuations, e.g. of the integrated current through the system, which
are Gaussian (this was done using a mapping to the ABC model). The main result of this paper
is to go beyond that work [43] and derive a non-Gaussian MFT, which in particular retains
the Poissonian nature of the noise arising from the random tumbling events.
Within this framework, they derive the large deviation function (LDF) associated to the integrated current.
Using an analogy with equilibrium phase transitions, they study in detail the dynamical
phase transition exhibited by this LDF. They also connect this phase transition to the rich
phase diagram of the model, which exhibits MIPS.

The present paper presents original, interesting and timely results for active particle systems.
In particular, I think that their study of the Poissonian noise should be relevant to study
other active systems. My only criticism is that the presentation of the results (and somehow
of the whole paper in general) could be made a bit more accessible to a wider audience (see also below
for other comments), not necessarily familiar with MIPS (which at the end is not really the main
topic of the paper). Once the authors take these remarks into account,
the paper can be published in SciPost.

Here is a list of comments that the authors might wish to consider (some of them are just typos or minor comments):

1) At the end of Section II, it would be useful if the authors could explain the phase diagram with
more details. As it is now, it is a bit cryptic to follow, in particular for readers who are not familiar with MIPS.

2) In Fig. 1, there are arrows to illustrate the mechanism 'D_0'. The second arrow (the rightmost one) is a bit misleading: what does it mean?

3) After Eq. (2), it would be useful to give a clear definition of $J_\rho$, $J_m$ and $K$.

4) In Eq. (5), I am not sure that ${\mathbb C}$ is the best notation since this usually denotes the ensembles of complex numbers...

5) Above (5): "account" --> "accounts"

6) Below (22): "with a conserved order parameters" --> "with conserved order parameters"

7) At the beginning of Section V, the authors should recall what $\ell_s$ is.

8) At the bottom of p. 6 "reminiscence" --> "reminiscent"

9) Below Eq. (30), "in low density phase" --> "in THE low density phase"

10) Below Eq. (30) it is not clear what is meant by "This dynamics are illustrated schematically in Fig. 5(B)"

11) Above A2): "transmutation reaction" --> "transmutation reactions"

---

## Round 1 · Referee Report · Anonymous · 2022-10-31

Report

This is an important and timely paper that builds on the authors substantial track record on large deviations and dynamical phase transitions. Here they study the full counting statistics of current fluctuations in a one-dimensional model of active matter (that of Kourbane-Houssene et al, ref.39). The most salient feature of (intrinsically non-equilibrium) active matter is the so-called motility induced phase separation (the non-homogeneous clustered phase akin to that of sticky spheres but in the absence of interactions, which occurs at high drive even in 1D). The model is a generalisation of a SEP with tumbling particles carrying charge which determines their motional asymmetry, with rates scaled with size appropriately to give a good diffusive limit of times and length.

There are two main results. The first one is the formulation of the MFT for this model (and by extension for similar active lattice systems). The presence of stochastic jumps due to tumbling events makes the MFT for the particle and "charge" currents non-Gaussian. The second result is the computation of the relevant LD functions, and in particular the observation of a "dynamical phase transition" (i.e. a transition in the optimal trajectories of rare events). The fact that these can be shown analytically is quite remarkable, making this paper an important starting point for the application of LD/MFT methods to this class of systems. An interesting result is that the LD transition coincides with the steady state MIPS one at the critical point of the model (as it occurs for LD phase transitions in systems with equilibrium phase transitions).

The paper is well written and mostly clear, and the results correct as far as I can check. One small issue is the use of "bias" which can be confusing: on the one hand it can refer to the asymmetry in the particle hopping (as after eq.22), while in LDs bias often refers to what is also called "tilting" (the exponential rewighting of the path probability to probe atypical events, controlled in this case by \Lambda). After fixing this and several typos the paper can be published.

---

## Round 2 · List of Changes

Dear Editor,
We wish to thank both referees for carefully studying the manuscript, for their positive feedback, and for the very useful comments and suggestions that we have fully implemented. Here is the list of changes made in the manuscript following the referee's comments. We hope that with these changes the manuscript can now be accepted for publication:
1) Referee (II) pointed out the confusing use of the word bias. We fully agree with him. Accordingly, we have modified the text just below equation (28) and avoided the use of "bias".
2) Referee (I) pointed out that the paper "could be made a bit more accessible to a wider audience, not necessarily familiar with MIPS". This was elaborated in his comments No (1), (2), and (3). We found these comments very useful and they helped us revise the text to address the accessibility issue: A) We have slightly reformulated the presentation of the dynamics of the model at the beginning of Sec. II and added a short explanation in the caption of Fig. 1. B) We have added a more detailed derivation and explanation of the fluctuating hydrodynamics. This additional text appears now between Eq. (1) and Eq. (7). C) We have added a detailed explanation for the MIPS and the phase diagram below Eq.(7). we also added a short explanation in the caption of Fig. (2).
3) Following Referee (I) comment No (4) we have changed the symbol \mathbbC -> \mathbfC.
4) Following his comment No (10) we have added a short explanation below Eq.(34) which now better links the text to Fig 5 (B).
5) All of the other typos and grammatical errors were fully implemented.

You are currently on this page

Resubmission scipost_202208_00013v2 on 7 November 2022

---

## Editorial Decision

published